# Zero-shot Topical Text Classification with LLMs - an Experimental Study

**Shai Gretz**,[*] **Alon Halfon**,[*] **Ilya Shnayderman, Orith Toledo-Ronen,**
**Artem Spector, Lena Dankin, Yannis Katsis, Ofir Arviv,**
**Yoav Katz, Noam Slonim, Liat Ein-Dor**

IBM Research
{avishaig,alonhal,ilyashn,oritht,artems,lenad,katz,noams,liate}@il.ibm.com,
{yannis.katsis,Ofir.Arviv}@ibm.com

## Abstract

Topical Text Classification (TTC) is an ancient, yet timely research area in natural language processing, with many practical applications. The recent dramatic advancements in large LMs raise the question of how well these models can perform in this task *in a zero-shot scenario*. Here, we share a first comprehensive study, comparing the zero-shot performance of a variety of LMs over $TTC^{23}$, a large benchmark collection of 23 publicly available TTC datasets, covering a wide range of domains and styles. In addition, we leverage this new TTC benchmark to create LMs that are specialized in TTC, by fine-tuning these LMs over a subset of the datasets and evaluating their performance over the remaining, held-out datasets. We show that the TTC-specialized LMs obtain the top performance on our benchmark, by a significant margin. Our code and model are made available for the community.[1] We hope that the results presented in this work will serve as a useful guide for practitioners interested in topical text classification.

## 1 Introduction

The recent emergence of Transformer–based Language Models (LMs) has led to significant breakthroughs in various NLP tasks (Brown et al., 2020; Chung et al., 2022). In particular, LMs have shown dramatic performance improvements in text classification (e.g., Zhang et al. (2023)), which is one of the most common use cases considered by NLP practitioners. Notably, these improvements are also present in the challenging setting of zero-shot classification, where no labeled data are available for the target categories (Yin et al., 2019). Manual collection of labeled data is known to be notoriously costly, complicated, and time consuming, representing a major blocker in the adoption of text classification solutions in practice. Hence, gaining a better understanding of the current performance of LMs in this zero-shot scenario is a timely issue with great practical importance.

Text classification is a relatively broad umbrella term, covering tasks such as (i) Sentiment Analysis (SA) in various forms (Liu, 2015); (ii) Style Detection - e.g., spam filtering (Cormack, 2008), authorship attribution, etc.; and (iii) Topical Text Classification (TTC), where the goal is to associate a text example with one or more topics out of a pre-defined set of topics, or categories (e.g., Lang (1995)).[2]

The zero-shot scenario is of crucial importance for the latter, TTC task. Specifically, for tasks such as SA, the categories are typically the same between different use cases – Positive, Negative, and Neutral. Hence, in principle, one can rely on existing SA labeled datasets to fine-tune the LM. In contrast, in TTC the list of categories is likely to change from one downstream task to another, hence relying on previously labeled datasets is not straightforward.

Thus, the main motivation for the present work relies on three intertwined pillars – (i) the special importance of the zero-shot setup for TTC; (ii) the wide range of practical TTC use cases – e.g., in domains such as customer-care, healthcare, and legal (Chalkidis et al., 2022); and (iii) the assumed potential of LMs to obtain strong zero-shot performance in TTC tasks.

Correspondingly, here we report the results of a comparative study - to the best of our knowledge, first of its kind - focused on the zero-shot TTC performance of a plethora of LMs over a large collection of TTC datasets. Specifically, we consider 23 datasets, across a variety of domains, styles and complexity levels, including news, legal contracts, chats and more.

---

[*]These authors equally contributed to this work.

[1] https://github.com/IBM/zero-shot-topical-text-classification

[2]We use the terms topics and categories interchangeably throughout the paper.

Moreover, while previous work used LMs for zero-shot TTC without specific tuning for this task, here, we suggest that specializing LMs in the TTC task can provide significant performance gains. To that end, we show that one can fine-tune a LM over existing TTC labeled datasets to significantly improve its performance over *new* TTC tasks, with categories never seen before. We show the value of this approach in zero-shot TTC by training the LM on a subset of the datasets, and evaluating its performance on the remaining, held-out datasets.

To summarize, the main contribution of this paper is three-fold: (1) We introduce $TTC^{23}$, a heterogeneous collection of 23 TTC datasets, that we propose as a new benchmark for this task; (2) we share the results of a comprehensive comparative study of the zero-shot performance of various LMs at different sizes over $TTC^{23}$, addressing both prediction performance and run-time considerations; and finally, (3) we show that $TTC^{23}$ can be leveraged to create models with significantly enhanced zero-shot performance in new TTC tasks.

## 2 Related Work

### 2.1 Zero-shot Topical Text Classification

Benchmarking TTC was highlighted in Zhang et al. (2015), introducing large-scale datasets, such as AG News, DBPedia and Yahoo Answers.

Zero-shot TTC has been mostly studied as part of a wider scope of zero-shot classification tasks in NLP. Yin et al. (2019) have shown the usefulness of leveraging LMs fine-tuned on NLI datasets for the purpose of zero-shot text classification in general, where topical text classification datasets were only one type of the evaluated tasks. Halder et al. (2020) introduced an approach named TARS, which unifies text classification tasks to a pairwise format, and is able to transfer knowledge from one dataset to another by leveraging the semantic relation between the text and the label in a zero or few-shot setting. Zhong et al. (2021) map text classification tasks into a pairwise question-answering format, where each class is given as a prompt in question format, with Yes/No labels. They also propose the concept of "meta-tuning", suggesting that teaching a model how to solve different tasks in a unified format can help it to better generalize to unseen tasks. Puri and Catanzaro (2019) proposed using generative models for zero-shot text classification. They formulated the text classification tasks, including TTC, as a multiple-choice

question answering problem, and used the GPT-2 model.

A recently introduced approach for creating zero-shot learners is instruction-tuning (Sanh et al., 2021; Chung et al., 2022; Longpre et al., 2023) in which zero-shot generalization is enabled by mapping natural language tasks into human-readable instructions and finetuning pretrained models with a multitask mixture of datasets covering a wide variety of tasks. A prominent example for this approach is Flan-T5, which was trained over 1800 tasks, including at least 4 datasets that are topical in nature. While the works mentioned above aimed to create a versatile zero-shot model that can perform well over a wide range of downstream tasks, our aim here is to explore whether honing-in on a narrower scope of downstream tasks, namely topical text classification, can enable us to improve the zero-shot performance on this type of task only.

A previous paper that focused specifically on benchmarking TTC is the work of Schopf et al. (2023), which conducted an evaluation of zero-shot TTC by comparing similarity-based and NLI-based models on 4 TTC datasets. In our work we expand the set of benchmark datasets and evaluated models, and further suggest to improve upon them via task-specific fine-tuning.

### 2.2 Evaluation of large LMs

There is a recent rise in attention to evaluation of large LMs in several contexts. Kocoń et al. (2023) and Chen et al. (2023) evaluated ChatGPT and GPT-3.5, respectively, on a range of NLP tasks, e.g., sentiment analysis, emotion recognition, and question answering, while Chalkidis (2023) did so for ChatGPT specifically on datasets in the legal domain. Zhang et al. (2023) evaluated Flan-UL2, Flan-T5-XXL, GPT3.5, and ChatGPT on tasks associated with sentiment analysis, finding that Flan-UL2 achieves comparable or even better results than GPT-3.5 or ChatGPT, despite being magnitudes smaller. Wadhwa et al. (2023) evaluated GPT-3 and Flan-T5-Large on relation extraction in a few-shot and supervised setting. Parikh et al. (2023) evaluated Flan-T5-XXL and GPT-3 on zero-shot intent detection on 4 benchmark datasets, finding that they achieve comparable results.

# 3 Approaches to Zero-shot Topical Text Classification

## 3.1 Existing Approaches

There are several approaches for dealing with zero-shot TTC. Here we briefly describe a collection of popular methods, highlighting their commonalities and differences. What all the approaches have in common is the key idea of casting different tasks onto a single meta-problem. Using this casting, a model that was trained to solve the meta-problem can also be used to solve other, unseen tasks. The three main meta-problems that have been suggested in the context of text classification are: Natural Language Inference (NLI), Question Answering (QA), and Instruction Tuning. Figure 1 shows how a TTC example is mapped onto each meta-problem.

**Natural Language Inference (NLI).** In this approach, initially proposed by Yin et al. (2019), pre-trained NLI encoder-only models are used as zero-shot text classifiers. These models are pre-trained using large manually labeled datasets for textual entailment, such as the MNLI dataset (Williams et al., 2018). The method operates by treating the text to be classified as the NLI premise and constructing a hypothesis for each candidate topic. The hypothesis template depends on the target task, where for TTC, the template is usually of the form "this text is about <topic>". The entailment and contradiction probabilities predicted by the NLI model are converted to label probabilities for this topic.

**Question-Answering (QA).** In this approach, different tasks are translated into yes/no questions. Zhong et al. (2021) created a large question answering meta-dataset from a collection of manually annotated datasets of different NLP classification tasks, all framed as question answering problems. Next, they used these data to train a binary model for yes/no question answering.

**Instruction Tuning.** Another approach for dealing with zero-shot text classification is by treating the classification task as a sequence generation task, where the target task is phrased as an input prompt and the text generated by the model is the predicted label. A well-known approach for improving the zero-shot performance of generative models is instruction tuning. This approach leverages the intuition that NLP tasks can be described via natural language instructions. Following this intuition, models are fine-tuned on a collection of tasks described via instructions. The instructions are fed to the model as input prompts.

## 3.2 Our Approach

Most available solutions for zero-shot TTC are based on models that were designed to solve a wider range of tasks. We propose to create models specialized at TTC by leveraging the large amount of publicly available topical datasets. A large collection of such datasets from various domains and styles is used to create a rich and diverse meta-dataset for TTC, that can be used to fine-tune existing zero-shot models, adapting them to the specific task of TTC.

The details on how we process the datasets and fine-tune different types of zero-shot models on them are presented next, in Sections 4 and 5.

# 4 The *TTC*[23] Benchmark

Our aim was to gather a diverse set of datasets containing multiple domains and styles that could serve as a training set for fine-tuning models, and also as a TTC evaluation benchmark. We collected datasets for TTC by searching on huggingface, kaggle, as well as in related papers, focusing on datasets where the input is between one and a few sentences, in English, and labeled in a multi-class or multi-label setting. The collection covers, among others, the following domains and styles: legal – both formal (e.g., LEDGAR) and informal (e.g., Legal Advice Reddit); finance – both from the news (Reuters) and tweets (Financial Tweets); News items – both headlines (News Category Classification Headline) and discussion threads (20 Newsgroups); and Chatbot queries – in banking (Banking77) and multi-domain (Massive).

Overall we collected 23 datasets, presented in Table 1.

## 4.1 Converting Datasets from Different Tasks

Our collection primarily includes datasets associated with detecting the main theme or topic in a given text. To enrich the diversity of the collection, we also considered datasets which were originally targeted at a separate task. For *Banking77*, *Clinc150*, and *Massive*, originally curated for the task of intent detection, we took the intent as the gold label; for *Argument Topic* and *Claim Stance Topic*, which contain arguments associated with debatable topics, we took the debatable topic as the gold label; and for *Contract NLI*, which was orig-

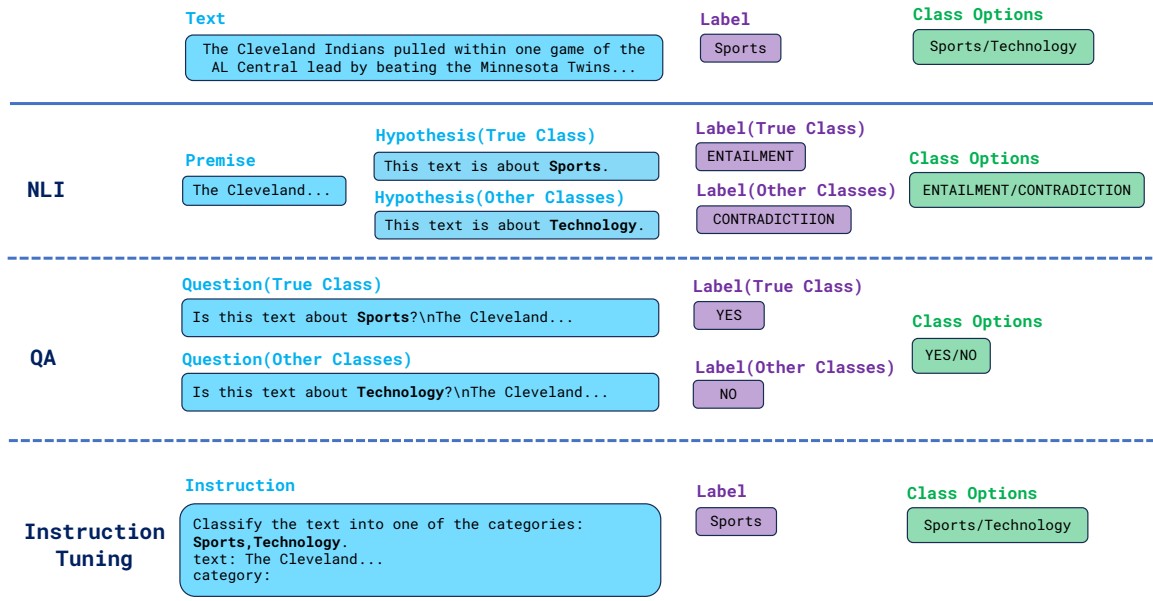

Figure 1: Framing topical text classification as each of the meta-problems representing the common zero-shot approaches.

| Dataset | Size | # Classes | Multi-label | Domain |
|---|---|---|---|---|
| 20 Newsgroups (Lang, 1995) | 19.0k | 20 | No | General |
| AG News (Zhang et al., 2015) | 142.6k | 4 | No | General |
| Argument Topic (Gretz et al., 2020) | 10.4k | 71 | No | General |
| Banking77 (Casanueva et al., 2020) | 14.5k | 77 | No | Finance/Banking |
| Claim Stance Topic (Bar-Haim et al., 2017) | 2.6k | 55 | No | General |
| Clinc150 (Larson et al., 2019) | 25.5k | 150 | No | General |
| Contract NLI (Koreeda and Manning, 2021) | 7.5k | 17 | Yes | Legal |
| CUAD (Hendrycks et al., 2021) | 8.8k | 37 | Yes | Legal |
| DBPedia (Zhang et al., 2015) | 650k | 14 | No | Legal |
| Eli5-category (Gao et al., 2021) | 91.6k | 10 | No | General |
| Financial Tweets (Jia, 2022) | 23.6k | 20 | No | Finance/Banking |
| HeadQA (Vilares and Gómez-Rodríguez, 2019) | 8.1k | 6 | No | Medical |
| Law Stack Exchange (Li et al., 2022) | 2.8k | 16 | No | Legal |
| LEDGAR (Chalkidis et al., 2022) | 88.5k | 99 | No | Legal |
| Legal Advice Reddit (Li et al., 2022) | 108.8k | 11 | No | Legal |
| Massive (FitzGerald et al., 2022; Bastianelli et al., 2020) | 18.5k | 60 | No | General |
| Medical Abstracts (Schopf et al., 2023) | 15.5k | 5 | No | Medical |
| News Category Classification Headline (Misra and Grover, 2021; Misra, 2022) | 229.5k | 40 | No | General |
| Reuters (Apté et al., 1994) | 13.1k | 20 | Yes | Finance/Banking |
| Sentivent (Jacobs, Gilles and Hoste, Veronique, 2022) | 7.5k | 18 | Yes | Finance/Banking |
| Unfair ToS (Chalkidis et al., 2022) | 11.6k | 8 | Yes | Legal |
| Xglue (Liang et al., 2020) | 130.0k | 10 | No | General |
| Yahoo Answers (Yin et al., 2019) | 145.4k | 10 | No | General |

Table 1: List of datasets in the $TTC^{23}$ benchmark collection.

inally curated for identifying entailment between premises and hypotheses in contracts, we took the category of the hypothesis as the gold label.

## 4.2 Data Formatting

To ease data processing, we converted all datasets to a unified format containing a single text column and a single label column, where each label contains zero or more categories (depending on the dataset). For datasets that originally did not contain a train-dev-test split, we created one.

## 4.3 Label Cleansing and Rephrasing

Zero-shot TTC relies on a good semantic representation of the topic. For it to be effective, this representation should be clear and grammatically sound,

and also consistent between different datasets. This led us to rephrase the categories in the following ways (see Appendix E for the full list of changes):

1. We rephrased categories that were in an unreadable or technical format (e.g., in 20 Newsgroups, we rephrased `alt.atheism` to `atheism`).

2. We rephrased categories that might not be grammatically correct when prepended with a prompt (e.g., in Clinc150, we rephrased `find_phone` to `finding a phone`).

3. We removed examples associated with categories that do not convey semantic information (e.g., topic `general` in LEDGAR).

4. For the Reuters dataset, many categories were either acronyms or required domain expertise to parse. Thus, we kept examples associated with the top-20 prevalent categories in the dataset, and rephrased these categories to make them more readable.

5. We performed further cleansing across all datasets to remove underscores and convert to lower case.

## 5 Experimental Setup

Next we describe the setup in which we evaluate the different methods for zero-shot TTC.

### 5.1 Datasets

We use the $TTC^{23}$ benchmark described in Section 4 containing 23 datasets – 18 multi-class and 5 multi-label – for training and evaluation.

### 5.2 Evaluating Fine-tuned Models on $TTC^{23}$

#### 5.2.1 Leave-one-fold-out

To evaluate fine-tuned models in a zero-shot setting, we split $TTC^{23}$ into $k$ folds and employ a leave-one-fold-out setup, where we train on $k$-1 folds and evaluate on the remaining fold.

We consider two fold-splits:

**In-domain.** In this split, different folds share datasets from the same domain. The motivation for this split is to evaluate how well a model performs when tested on a similar domain that was present in training. We split to 3 folds containing 8, 8, and 7 datasets, respectively. Unless stated otherwise, this fold-split is used for all experiments. The folds are presented in Appendix A.

**Out-of-domain.** In this split, different folds do not share datasets from the same domain. The motivation for this split is to evaluate how well a model performs when tested on a domain not present in training. We determined for each dataset its domain manually, ending up with 4 folds, as presented in Table 1.

#### 5.2.2 Building a Unified Dataset

When fine-tuning a model in the leave-one-fold-out setup, we create merged train and dev sets. The merged train set is comprised of the train sets of all datasets in the folds we train on (and likewise, the merged dev set). For the train set, we also sample at most 100 examples from each category to prevent over-dominance of highly populated categories. For evaluation, we consider the test sets of all datasets in the held-out fold separately, and they are not sampled.

For building the merged train/dev sets for NLI-based models, we convert the positive examples to a pairwise format, where each example contains the text input and the hypothesis template "This text is about <topic>.", separated by `[SEP]`, with an `ENTAILMENT` label. In case the example contains multiple positive topics (in a multi-label setting), we create multiple pairwise positive examples, one for each positive topic.

For each positive pairwise example we add a negative pairwise example whose topic is selected at random from the categories of this dataset (excluding the positive topics of this example), with a `CONTRADICTION` label.

For instruction-tuned models, we similarly merge the datasets in the training folds. The input for each text is comprised of the instruction, the candidate categories, and the text, and the output is the target category. If the example originally contains more than one category, we choose one randomly. Examples with no categories are ignored in training. We use the instruction in Figure 1.

### 5.3 Inference

**Encoder-only Models.** For all encoder-only models we run a pairwise inference of each example with all candidate categories.

**Encoder-Decoder Models.** For instruntion-tuned models, when evaluating multi-class datasets we construct the same instruction that contains the task description and all candidate categories, and ask the model to generate the predicted category.

Classification tasks are a special case of a generation task where the generated tokens are expected to come from a pre-defined list of categories. Thus, we implemented a logits processor that restricts the calculation of tokens to take into account only the tokens that are permitted according to the candidate categories. This avoids generating texts which cannot be exactly matched to any category and improves the performance. For example, if there are two candidate categories, "world news" and "tv and film", and the first generated token is "world", the second token must be "news".

Multi-label datasets have been relatively less explored in the context of using instruction-tuned models for classification. We propose to frame the classification as a binary classification task. Instead of asking the model to generate a single category name out of a pre-defined list of candidates (as in the multi-class setting), we present the model with each candidate category separately, along side its negation (e.g., *sports, not sports*), and ask the model to choose between the two. The score for each <text, category> pair is the probability given by the model for generating that category name.

## 5.4 Models

We consider the following off-the-shelf models:

**S-BERT** (Reimers and Gurevych, 2019). In addition to the approaches in Section 3, we also evaluate S-BERT. We use the all-mpnet-base-v2 model.

**QA-based.** We use the RoBERTa-Large-QA model released in Zhong et al. (2021).[3] For each inferred pair, we calculate the probability of the label "Yes". We use a single label description (question), *'Is this text about <topic>?'*, which is one of the manually annotated label descriptions used by Zhong et al. (2021), and matches the TTC task. Three of the $TTC^{23}$ datasets were part of the training set of this model, thus we also report results on a subset of $TTC^{23}$ that excludes them.

**NLI-based.** We use RoBERTa-large-NLI (Liu et al., 2019)[4] and DeBERTa-large-NLI (He et al., 2021).[5] For each inferred pair, we calculate the probability of the label ENTAILMENT divided by the probability of the label CONTRADICTION.

[3]https://huggingface.co/ruiqi-zhong/roberta-large-meta-tuning-test

[4]https://huggingface.co/roberta-large-mnli

[5]https://huggingface.co/MoritzLaurer/DeBERTa-v3-large-mnli-fever-anli-ling-wanli

**Instruction-tuned models.** We evaluate Flan-T5-Large/XL/XXL. Four of the datasets in $TTC^{23}$ were part of Flan's training, thus we also report results on a subset of $TTC^{23}$ that excludes them. Given that datasets may contain a large set of candidate categories, increasing the size of the prompt, we use a sequence length of $2048$. We use a greedy decoding method, since our aim is to generate the tokens representing the most probable class.

We also evaluate models fine-tuned over $TTC^{23}$ in the leave-one-fold-out setup described above. Implementation details can be found in Appendix B. We consider the following fine-tuned models:

1. NLI-based FT. We fine-tune RoBERTa-large-NLI and DeBERTa-large-NLI.

2. Instruction-tuned FT. We fine-tune Flan-T5-XXL. In order to be able to fine-tune this model, we use the efficient fine-tuning approach of Low Rank Adapters (LoRA) (Hu et al., 2021). We fine-tune the model for 3 epochs and use the dev set for early stopping.

### 5.4.1 Decoder-only Models

Recent advancements exhibited in GPT-4 and ChatGPT suggest they could be suitable for TTC as well. However, there are several issues associated with evaluating these models. First, assessing the zero-shot performance of the GPT family is challenging due to its undisclosed training data, potentially encompassing supervised training from our evaluation datasets. Second, given its paid nature, extensive inference at the scale presented in this work is not only costly but also non-reproducible should the service evolve in the future. Thus, we excluded them from this paper.

## 5.5 Metrics

**Multi-class.** For multi-class datasets, we take the highest scoring predicted category per example, and report macro-averaged f1 over all categories.

**Multi-label.** For multi-label datasets, to avoid sensitivity to any decision threshold, we report macro-average AUC-ROC over all categories.

We report the average of 3 seeds.

## 6 Results and Analysis

We present results averaged on all datasets in Table 2 and full results in Tables 4 and 5 in the Appendix.

| Model | Model Type | #Params | MC | MC* | MC** | ML |
|---|---|---|---|---|---|---|
| **Off-the-shelf Models** | | | | | | |
| S-BERT | | 110M | 52.66 | 51.89 | 54.15 | 85.49 |
| RoBERTa-Large-QA | QA-based | 355M | - | - | 54.21 | 88.12 |
| RoBERTa-Large-NLI | NLI-based | 355M | 51.05 | 47.89 | 51.26 | 82.97 |
| DeBERTa-Large-NLI | NLI-based | 435M | 54.40 | 51.08 | 55.15 | 88.26 |
| Flan-T5-Large | Instruction-tuned | 770M | - | 54.89 | - | 86.57 |
| Flan-T5-XL | Instruction-tuned | 3B | - | 61.39 | - | 89.47 |
| Flan-T5-XXL | Instruction-tuned | 11B | - | 64.79 | - | 89.72 |
| **Models fine-tuned on $TTC^{23}$** | | | | | | |
| RoBERTa-Large-NLI FT | NLI-based | 355M | 58.59 | 57.51 | 59.34 | 90.64 |
| DeBERTa-Large-NLI FT | NLI-based | 435M | 64.00 | 63.14 | 65.36 | **93.19** |
| Flan-T5-XXL FT | Instruction-tuned | 11B | - | **67.32** | - | 89.56 |

Table 2: TTC results over 18 multi-class and 5 multi-label datasets in $TTC^{23}$. Top: off-the-shelf zero-shot models. Bottom: fine-tuned models over $TTC^{23}$ in leave-one-fold-out. MC stands for multi-class datasets, and ML for multi-label datasets. MC* are the subset of 14 datasets not included in Flan models training. MC** are the subset of 15 datasets not included in the QA model training. For MC/*/** we report macro-f1 and for ML macro-AUC-ROC, averaged over all respective datasets.

## 6.1 Off-the-shelf Models

Considering the top part of Table 2, Flan-T5-XXL model is clearly the best zero-shot model for TTC. As expected, the performance of Flan-based models increases with model size (see MC* column). Note, that QA and Flan models were trained on multiple tasks (including TTC ones), making them a stronger baseline compared to the NLI-based models which were trained on a single and somewhat different entailment task. It is also worth noting that S-BERT is competitive with off-the-shelf encoder-only models, representing a much faster cost-effective alternative when resources are limited.[6]

## 6.2 Impact of Fine-Tuning on $TTC^{23}$

The results of models fine-tuned over $TTC^{23}$ in leave-one-fold-out setup are presented at the bottom of Table 2. Remarkably, fine-tuning RoBERTa and DeBERTa models significantly improves their macro-f1 performance on MC datasets by 8 and 10 points, respectively; and by 10 and 12 points, respectively, over the MC* datasets. Fine-tuning Flan-T5-XXL yields a more moderate, yet significant improvement of 2.5 f1 points on the MC* datasets, leading to the top zero-shot performance in our MC experiments with an impressive macro-

averaged f1 score of 67.32. As can be seen in Table 4, it is the best model for 10/14 of the MC* datasets. Interestingly, DeBERTa-Large-NLI FT outperforms the much larger Flan-T5-XL, which further highlights the advantage of the task-specific training.

Focusing on encoder-only models, and the MC** column, fine-tuning on $TTC^{23}$ yields better performance compared to the QA-based model, that uses a different training scheme.

## 6.3 Comparison to Leave-one-domain-out

We compare the results of our in-domain setup to an out-of-domain setup, where the evaluated datasets in each fold come from a domain not present in the training data, using DeBERTa-Large-NLI FT. This enables to quantify the impact of domain similarity between train and test on model performance. The results are in Table 3. When moving to an out-of-domain setup macro-f1 and macro-AUC-ROC drop by 1-2 points, as expected. The out-of-domain setup is more challenging, but not necessarily more realistic. Given that we envision a model trained on hundreds of classes and multiple domains, it might be reasonable to assume that in real-world applications this model will be tested over datasets it is already somewhat familiar with.

## 6.4 Does Category Similarity Help?

As an additional measure for evaluating the impact of training data on model performance, we explore

---

[6]In addition to the models presented here, we experimented with the Llama family of models. Our trials in zero-shot classification with them on $TTC^{23}$ yielded significantly inferior results, leading to its exclusion.

whether high performance on categories in the test fold is correlated with the occurrence of semantically similar categories in the train set. To that end, for each category in a test fold we find the score of its most similar category in the train folds using S-BERT. The correlation between these scores and the respective f1 scores in the DeBERTa-Large-NLI FT run is insignificant (-0.03). Furthermore, in an anecdotal inspection, we observe relatively low performance for categories in the test fold even though the exact same category name was included in the train folds. E.g., the category `greeting` appears both in *Clinc150* and *Massive*, however the f1 results for this category are lower compared to other categories. Thus, overall it seems that the semantic similarity of a category name to another category included in the training data has no significant influence over performance, possibly due to the magnitude and diversity of the training data we consider.

### 6.5 Run-time Analysis

When choosing which model to run, a relevant aspect for a practitioner is run-time. To this end, we analyze the run-time of the different models. Each of the models we consider uses a different technique for running predictions: S-BERT calculates the embeddings of the input texts and categories separately and calculates their cosine similarity. Flan-T5-XXL concatenates all the candidate categories to a single prompt with additional tokens for instruction, and runs one inference on each input text – thus, the number of categories directly impacts the prompt length. DeBERTa-Large-NLI prepends a short template to each text, and runs inference for each category separately. Thus, the number of categories directly impacts the number of inferences per input text.

Inference over large datasets naturally takes more time than inference on smaller datasets. To compare run-time between datasets of different sizes, we measure the throughput using the Kchar/s (kilo characters per second) metric, dividing the total length of the dataset input by the run-time.

We use an A100 GPU with 40GB of memory. For each method, we pick the inference batch size that allows minimal run-time as measured in seconds. We use fp16 quantization for both DeBERTa-Large-NLI and Flan-T5-XXL.

The average throughput of S-BERT, DeBERTa-Large-NLI, and Flan-T5-XXL is 118.73, 21.67,

| Model | MC | ML |
|---|---|---|
| DeBERTa-Large-NLI FT | 64.00 | 93.19 |
| DeBERTa-Large-NLI FT (OOD) | 61.83 | 92.04 |

Table 3: TTC over 18 multi-class and 5 multi-label datasets in $TTC^{23}$, comparing in-domain (top) to out-of-domain (bottom) fold-splits.

and 8.02 Kchar/s respectively. Full analysis is available in Table 6 in the Appendix. Next, we share a few observations that emerge from this analysis.

There is large variance between different datasets, as text length and number of categories impact the throughput. For example, Flan-T5-XXL's throughput on Clinc150 is 0.43 Kchars/s compared to 22.1 Kchars/s on Legal Advice Reddit. Similar variance is seen in other models.

For most datasets, the throughput of DeBERTa-Large-NLI is higher, i.e., better than that of Flan-T5-XXL, despite making more inference calls per text. For example, in Xglue the throughput of DeBERTa-Large-NLI is more than 4 times greater than with Flan-T5-XXL. The only exception to this is LEDGAR, which contains long texts and a large number of categories. For this dataset, the throughput of Flan-T5-XXL is two times larger. This is presumably because Flan-T5-XXL processes the long text only once, and concatenates the candidate categories to a single inference, while DeBERTa-Large-NLI infers the text repeatedly for each candidate category. Coupled with the results discussed in Section 6.2, this analysis highlights the run-time vs. quality trade-off, as the superior quality of Flan-T5-XXL comes at the cost of longer run-time.[7]

For S-BERT, the effect of calculating category embeddings and cosine similarity is negligible, and the model does not use any prompt. Aside from the fact that it is the smallest model, S-BERT naturally has the best throughput across all datasets, at the cost of lower f1.

## 7 Conclusions

TTC is an ancient problem in computer science research, dating back to 1961 (Maron, 1961). The recent advancements in large LMs led us to explore how well they perform on this problem in a zero-shot setting. In this paper we introduce a comprehensive evaluation of existing approaches to zero-shot TTC over a diverse set of datasets, $TTC^{23}$. Our results indicate that large LMs indeed

---

[7]Note, Flan-T5-XXL also has relatively high hardware constraints, though we did not analyze this aspect in detail.

exhibit a significant improvement over the smaller encoder-only models that were the main focus on previous studies on zero-shot TTC (e.g., (Yin et al., 2019; Schopf et al., 2023)).

Furthermore, we show that fine-tuning RoBERTa and DeBERTa models as well as Flan-T5-XXL over existing TTC datasets can significantly boost their respective zero-shot performance on *new* TTC datasets, with category names not included in the training data. In other words, even LLMs that represent very strong baselines, can be further significantly improved with additional fine-tuning focused on the target task. Our run-time analysis has shown that this superior performance comes with a cost of additional run-time, something the practitioner needs to take into account.

Our fine-tuned Flan-T5-XXL model obtained very impressive zero-shot performance - macro-averaged f1 of 67.32 when averaging across 14 multi-class datasets. Nonetheless, these results are typically still not satisfactory in practice, and we expect that a traditional process of collecting labeled data for the target task under consideration to further fine-tune the model would be required. That said, a strong zero-shot TTC model can serve as an excellent starting point to perform domain adaptation on a target dataset, e.g., by means of self-training (Gera et al., 2022) or active learning (Ein-Dor et al., 2020; Shnarch et al., 2022), as we plan to explore in future work.

## 8   Ethics and Broader Impact

This paper is submitted in the wake of a tragic terrorist attack perpetrated by Hamas, which has left our nation profoundly devastated. On October 7, 2023, thousands of Hamas terrorists infiltrated the Israeli border, launching a brutal assault on 22 Israeli villages. They methodically moved from home to home brutally torturing and murdering more than a thousand innocent lives, spanning from infants to the elderly. In addition to this horrifying loss of life, hundreds of civilians were abducted and taken to Gaza. The families of these abductees have been left in agonizing uncertainty, as no information, not even the status of their loved ones, has been disclosed by Hamas.

We fervently call for the immediate release of all those who have been taken hostage and urge the academic community to unite in condemnation of these unspeakable atrocities committed by Hamas.

We call all to join us in advocating for the prompt and safe return of the abductees, as we stand together in the pursuit of justice and peace.

## 9   Limitations

The approach taken in this work has a few limitations:

1. For reasons detailed in Section 5.4.1, we do not present an evaluation of decoder-only models such as GPT-4 or ChatGPT.

2. For Flan-based models, it could be that with better prompt engineering, e.g., including trying few-shot prompts, one could improve their performance.

3. Our approach for classification of multi-label datasets with the generative models is a rough attempt to adapt these models to this setup. It could be that other approaches for multi-label classification could be utilized, e.g., by asking the model to generate zero or more categories.

4. We analyze the potential impact of the training data on model performance in Sections 6.3 and 6.4. However, perhaps there are other, more subtle ways in which the training data impacts the performance which are not directly evaluated in this work.

5. We do not attempt to fine-tune the QA-based model, which could have benefited from it as well.

6. We consider datasets containing sentences, paragraphs, or short sections. We do not test our approach on document-level inputs, though in theory this should be feasible.

7. This type of work entails several degrees of freedom – what templates to use, which datasets to evaluate on, what hyper-parameters to tune, which runtime measures to consider, etc. We attempted to prioritize them in order to cover the issues we consider most important. However, dedicating more time and effort to these items could have yielded additional insights.

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

## A In-domain Folds

In the in-domain fold-split, these are the folds we considered:

Fold1: Reuters, Claim Stance Topic, Unfair ToS, HeadQA, Banking77, AG News, Yahoo Answers.

Fold2: Argument Topic, CUAD, DBPedia, News Category Classification Headline, Eli5-category, Financial Tweets, Law Stack Exchange, Massive.

Fold3: Clinc150, 20 Newsgroups, Contract NLI, LEDGAR, Legal Advice Reddit, Medical Abstracts, Xglue, Sentivent.

## B Model Implementation

For DeBERTa-based models, we fine-tune for 3 epochs without early-stopping using a learning-rate of 5e-6, a batch-size of 32, and a maximum sequence length of 256.

For RoBERTa-based models, we use a maximum sequence length of 128, which is the limit of the RoBERTa-Large-QA released model. For the RoBERTa-Large-NLI model, we fine-tune for 3 epochs without early stopping using a learning-rate of 1e-5, and a batch-size of 32.

When fine-tuning Flan-T5-XXL with LoRA, we adapt only the q (query) and v (value) projections in the transformer, and set rank=16, alpha=32 and dropout=0.05. We set the learning rate to 3e-5, the batch size to 16, the gradient accumulation steps to 16, the input sequence length to 512, the warmup steps to 2 and use a paged Adamw optimizer.

## C Full MC Results

Tables 4 and 5 present the full results of our evaluation over TTC[23].

## D Runtime Results

Table 6 presents the runtime analysis over all multiclass datasets of TTC[23].

## E Rephrasing Labels

| Dataset | S-BERT | RoBERTa-Large-NLI | RoBERTa-Large-NLI FT | RoBERTa-Large-QA | DeBERTa-Large-NLI | DeBERTa-Large-NLI FT | Flan-Large | Flan-XL | Flan-XXL | Flan-XXL FT |
|---|---|---|---|---|---|---|---|---|---|---|
| 20 News-group | 59.68 | 50.42 | 53.53 | 41.16 | 56.63 | 66.87 | 57.1 | 62.33 | 62.28 | **72.2** |
| AG News | 59.85 | 70.63 | 76.69 | | 68.67 | **79.55** | | | | |
| Argument Topic | 95.46 | 84.75 | 94.7 | 76.38 | 89.38 | 96.65 | 88.64 | 94.49 | 95.73 | **97.83** |
| Banking77 | 60.42 | 36.91 | 56.99 | 48 | 49.96 | 66.3 | 55.6 | 62.76 | 66.59 | **68.89** |
| Claim Stance Topic | 69.82 | 42.98 | 75.75 | 50.35 | 56.94 | 78.75 | 73.23 | 75.99 | 82.21 | **87.28** |
| Clinc150 | 69.15 | 51.87 | 71.98 | 54.1 | 54.73 | 79.06 | 61.4 | 72.92 | 79.11 | **83.47** |
| DBPedia | 68.88 | 82.56 | 82.54 | 84.1 | **92.76** | 90.18 | | | | |
| Eli5-category | 45.36 | 48.89 | 39.47 | 50.97 | 53.06 | 50.64 | 50.3 | 49.25 | 56.34 | **57.11** |
| Financial Tweets | 30.21 | 32.32 | 46.61 | 40.82 | 27.94 | 48.68 | 46.97 | **53.82** | 52 | 49.48 |
| HeadQA | 40.94 | 41.35 | 30.97 | **45.83** | 44.24 | 36.81 | | | | |
| Law Stack Exchange | 45.73 | 54.03 | 57.54 | 54.86 | 57.66 | 59.94 | 52.35 | 52.78 | 58.17 | **62.94** |
| LEDGAR | 23.28 | 33.77 | 46.44 | 37.61 | 18.5 | 48.99 | 31.89 | 44.73 | 47.73 | **55.86** |
| Legal Advice Reddit | 56.78 | 63.69 | 66.69 | 68.76 | 67.45 | 72.21 | 62.56 | 67.94 | 75.59 | **77.99** |
| Massive | 50.61 | 48.78 | 58.31 | 50.9 | 54.22 | 66.61 | 46.22 | 61.27 | 64.82 | **68.54** |
| Medical Abstracts | 58.4 | 53.39 | 53.74 | 54.34 | 53.6 | 59.13 | 54.04 | 62.48 | **63.71** | 60.69 |
| News Category Classification Headline | 24.06 | 25.5 | 28.57 | | 24.77 | 30.53 | 29.34 | 33.04 | **37.64** | 36.91 |
| Xglue | 37.56 | 43.12 | 54.82 | 54.97 | 50.24 | 59.58 | 58.83 | **65.67** | 65.13 | 63.25 |
| Yahoo Answers | 51.64 | 54.02 | 59.23 | | 58.42 | **61.59** | | | | |

Table 4: Full results of all models on 18 MC TTC[23] datasets.

| Dataset | S-BERT | RoBERTa-Large-NLI | RoBERTa-Large-NLI FT | RoBERTa-Large-QA | DeBERTa-Large-NLI | DeBERTa-Large-NLI FT | Flan-T5-Large | Flan-T5-XL | Flan-T5-XXL | Flan-T5-XXL FT |
|---|---|---|---|---|---|---|---|---|---|---|
| Contract NLI | 75.55 | 71.8 | 80.33 | 76.31 | 74.95 | **87.10** | 75.85 | 76.66 | 75.42 | 76.17 |
| CUAD | 88.49 | 86.04 | 94.65 | 92.10 | 88.96 | **96.71** | 92.27 | 93.66 | 92.46 | 92.39 |
| Reuters | 93.58 | 88.2 | 97.35 | 97.26 | 96.12 | **98.31** | 87.83 | 93.62 | 95.2 | 94.7 |
| Sentivent | 78.75 | 80.75 | 83.79 | 86.18 | 85.52 | 85.71 | 81.88 | 85.32 | **87.17** | 86.29 |
| Unfair ToS | 91.06 | 88.05 | 97.08 | 88.73 | 95.76 | 98.10 | 95.04 | 98.08 | **98.36** | 98.23 |

Table 5: Full results of all models on 5 ML TTC[23] datasets.

| Dataset | # Categories | Average Text Length | S-BERT | Deberta-Large-NLI | Flan-T5-XXL |
|---|---|---|---|---|---|
| 20 Newsgroups | 20 | 1925.64 | 274.96 | 21.94 | 15.19 |
| AG News | 4 | 232.49 | 87.12 | 54.33 | 13.78 |
| Argument Topic | 71 | 116.73 | 64.91 | 2.97 | 1.72 |
| Banking77 | 77 | 59.04 | 29.16 | 1.45 | 0.85 |
| Claim Stance Topic | 55 | 79.47 | 43.13 | 2.68 | 1.00 |
| Clinc150 | 150 | 38.02 | 25.82 | 0.57 | 0.43 |
| DBPedia | 14 | 280.08 | 104.60 | 17.60 | 5.87 |
| Eli5-category | 10 | 82.88 | 44.21 | 15.21 | 4.30 |
| Financial Tweets | 20 | 135.74 | 59.96 | 8.15 | 3.10 |
| HeadQA | 6 | 116.80 | 39.22 | 28.81 | 4.05 |
| Law Stack Exchange | 16 | 849.40 | 129.22 | 20.82 | 18.33 |
| LEDGAR | 99 | 715.78 | 117.31 | 3.91 | 8.37 |
| Legal Advice Reddit | 11 | 1292.77 | 202.30 | 37.16 | 22.51 |
| Massive | 60 | 31.36 | 22.25 | 1.24 | 0.68 |
| Medical Abstracts | 5 | 1185.45 | 182.09 | 72.57 | 17.66 |
| News Category Classification Headline | 40 | 59.09 | 38.30 | 3.15 | 1.39 |
| Xglue | 10 | 3971.01 | 589.28 | 71.59 | 16.42 |
| Yahoo Answers | 10 | 508.63 | 83.47 | 26.05 | 8.84 |

Table 6: Throughput in Kchars/sec on a Tesla A100 with 40Gb.

| dataset | class name | rephrased class name | rephrased type |
|---|---|---|---|
| claim stance topic | the one-child policy of the republic of China | the one-child policy of the republic of china | lower case |
| claim stance topic | make physical education compulsory | physical education | motion to topic |
| claim stance topic | subsidize the growing of tobacco | the growing of tobacco | motion to topic |
| claim stance topic | American Jobs Act | american jobs act | lower case |
| claim stance topic | all nations have a right to nuclear weapons | all nations a right to nuclear weapons | motion to topic |
| claim stance topic | subsidize poor communities | poor communities | motion to topic |
| claim stance topic | institute a mandatory retirement age | a mandatory retirement age | motion to topic |
| claim stance topic | re-engage with Myanmar | re-engage with myanmar | lower case |
| claim stance topic | build the Keystone XL pipeline | the keystone xl pipeline | motion to topic |
| claim stance topic | Israel's 2008-2009 military operations against Gaza | israel's 2008-2009 military operations against gaza | lower case |
| claim stance topic | build high rises for housing | high rises for housing | motion to topic |
| claim stance topic | the blockade of Gaza | the blockade of gaza | lower case |
| claim stance topic | Holocaust denial | holocaust denial | lower case |
| claim stance topic | the creation of private universities in the UK | the creation of private universities in the uk | lower case |
| claim stance topic | ASEAN | asean | lower case |
| claim stance topic | implement playoffs in collegiate level American football | implement playoffs in collegiate level american football | lower case |
| claim stance topic | unleash the free market | the free market | motion to topic |
| claim stance topic | have children | children | motion to topic |
| claim stance topic | build hydroelectric dams | hydroelectric dams | motion to topic |

Table 7: Label cleaning.

| dataset | class name | rephrased class name | rephrased type |
|---|---|---|---|
| argument topic | Intelligence tests bring more harm than good | intelligence tests | motion to topic |
| argument topic | Surrogacy should be banned | surrogacy | motion to topic |
| argument topic | We should ban cosmetic surgery | cosmetic surgery | motion to topic |
| argument topic | We should abolish capital punishment | capital punishment | motion to topic |
| argument topic | We should ban cosmetic surgery for minors | cosmetic surgery for minors | motion to topic |
| argument topic | We should ban human cloning | human cloning | motion to topic |
| argument topic | We should limit executive compensation | executive compensation | motion to topic |
| argument topic | We should ban naturopathy | naturopathy | motion to topic |
| argument topic | We should abolish the three-strikes laws | the three-strikes laws | motion to topic |
| argument topic | We should legalize organ trade | organ trade | motion to topic |
| argument topic | We should prohibit flag burning | flag burning | motion to topic |
| argument topic | We should adopt gender-neutral language | gender-neutral language | motion to topic |
| argument topic | We should subsidize Wikipedia | wikipedia | motion to topic |
| argument topic | We should legalize cannabis | cannabis | motion to topic |
| argument topic | We should introduce compulsory voting | compulsory voting | motion to topic |
| argument topic | We should limit judicial activism | judicial activism | motion to topic |
| argument topic | We should adopt a multi-party system | multi-party system | motion to topic |
| argument topic | We should adopt libertarianism | libertarianism | motion to topic |
| argument topic | Homeschooling should be banned | homeschooling | motion to topic |

Table 8: Label cleaning.

| dataset | class name | rephrased class name | rephrased type |
|---|---|---|---|
| argument topic | We should subsidize student loans | student loans | motion to topic |
| argument topic | We should subsidize stay-at-home dads | stay-at-home dads | motion to topic |
| argument topic | Payday loans should be banned | payday loans | motion to topic |
| argument topic | Assisted suicide should be a criminal offence | assisted suicide | motion to topic |
| argument topic | Holocaust denial should be a criminal offence | holocaust denial | motion to topic |
| argument topic | Social media brings more harm than good | social media | motion to topic |
| argument topic | We should ban private military companies | private military companies | motion to topic |
| argument topic | The use of public defenders should be mandatory | the use of public defenders | motion to topic |
| argument topic | We should abandon the use of school uniform | the use of school uniform | motion to topic |
| argument topic | Foster care brings more harm than good | foster care | motion to topic |
| argument topic | We should ban targeted killing | targeted killing | motion to topic |
| argument topic | We should fight for the abolition of nuclear weapons | the abolition of nuclear weapons | motion to topic |
| argument topic | We should ban algorithmic trading | algorithmic trading | motion to topic |
| argument topic | We should ban whaling | whaling | motion to topic |
| argument topic | The vow of celibacy should be abandoned | the vow of celibacy | motion to topic |
| argument topic | We should legalize prostitution | prostitution | motion to topic |
| argument topic | We should adopt a zero-tolerance policy in schools | zero-tolerance policy in schools | motion to topic |
| argument topic | We should abolish zoos | zoos | motion to topic |
| argument topic | We should abandon marriage | marriage | motion to topic |

Table 9: Label cleaning.

| dataset | class name | rephrased class name | rephrased type |
|---|---|---|---|
| argument topic | We should abandon television | television | motion to topic |
| argument topic | We should abolish intellectual property rights | intellectual property rights | motion to topic |
| argument topic | We should end mandatory retirement | retirement | motion to topic |
| argument topic | We should abolish the right to keep and bear arms | the right to keep and bear arms | motion to topic |
| argument topic | Blockade of the Gaza Strip should be ended | blockade of the gaza strip | motion to topic |
| argument topic | We should subsidize vocational education | vocational education | motion to topic |
| argument topic | We should stop the development of autonomous cars | the development of autonomous cars | motion to topic |
| argument topic | We should ban the use of child actors | the use of child actors | motion to topic |
| argument topic | We should adopt an austerity regime | austerity regime | motion to topic |
| argument topic | We should adopt atheism | atheism | motion to topic |
| argument topic | We should end affirmative action | affirmative action | motion to topic |
| argument topic | We should prohibit women in combat | women in combat | motion to topic |
| argument topic | We should ban the Church of Scientology | the church of scientology | motion to topic |
| argument topic | We should legalize sex selection | sex selection | motion to topic |
| argument topic | We should prohibit school prayer | school prayer | motion to topic |
| argument topic | Entrapment should be legalized | entrapment legalized | motion to topic |
| argument topic | We should close Guantanamo Bay detention camp | guantanamo bay detention camp | motion to topic |
| argument topic | We should ban factory farming | factory farming | motion to topic |
| argument topic | We should end racial profiling | racial profiling | motion to topic |

Table 10: Label cleaning.

| dataset | class name | rephrased class name | rephrased type |
|---|---|---|---|
| argument topic | We should ban telemarketing | telemarketing | motion to topic |
| argument topic | Homeopathy brings more harm than good | homeopathy | motion to topic |
| argument topic | We should ban missionary work | missionary work | motion to topic |
| argument topic | We should cancel pride parades | cancel pride parades | motion to topic |
| argument topic | We should legalize polygamy | polygamy | motion to topic |
| argument topic | We should abolish safe spaces | safe spaces | motion to topic |
| argument topic | We should oppose collectivism | collectivism | motion to topic |
| argument topic | We should fight urbanization | fight urbanization | motion to topic |
| argument topic | We should ban fast food | fast food | motion to topic |
| argument topic | We should subsidize embryonic stem cell research | embryonic stem cell research | motion to topic |
| argument topic | We should subsidize space exploration | space exploration | motion to topic |
| argument topic | We should end the use of economic sanctions | the use of economic sanctions | motion to topic |
| argument topic | We should abolish the Olympic Games | the olympic games | motion to topic |
| argument topic | We should subsidize journalism | journalism | motion to topic |
| clinc150 | translate | translation | grammatical |
| clinc150 | meaning_of_life | meaning of life | cleaning |
| clinc150 | insurance_change | insurance change | cleaning |
| clinc150 | find_phone | finding a phone | grammatical |
| clinc150 | travel_alert | travel alert | cleaning |
| clinc150 | pto_request | pto request | cleaning |
| clinc150 | improve_credit_score | improving credit score | grammatical |
| clinc150 | fun_fact | a fun fact | grammatical |
| clinc150 | change_language | changing language | grammatical |
| clinc150 | replacement_card_duration | replacement card duration | cleaning |
| clinc150 | application_status | application status | cleaning |

Table 11: Label cleaning.

| dataset | class name | rephrased class name | rephrased type |
|---|---|---|---|
| clinc150 | flight_status | flight status | cleaning |
| clinc150 | flip_coin | fliping a coin | grammatical |
| clinc150 | change_user_name | changing user name | grammatical |
| clinc150 | where_are_you_from | where you are from | grammatical |
| clinc150 | shopping_list_update | shopping list update | cleaning |
| clinc150 | what_can_i_ask_you | what i can ask you | grammatical |
| clinc150 | maybe | not being sure | grammatical |
| clinc150 | oil_change_how | how to change oil | grammatical |
| clinc150 | restaurant_reservation | restaurant reservation | cleaning |
| clinc150 | confirm_reservation | confirming reservation | grammatical |
| clinc150 | freeze_account | freezing account | grammatical |
| clinc150 | rollover_401k | rollover 401k | cleaning |
| clinc150 | who_made_you | who made you | cleaning |
| clinc150 | user_name | user name | cleaning |
| clinc150 | next_song | next song | cleaning |
| clinc150 | restaurant_suggestion | a restaurant suggestion | grammatical |
| clinc150 | rewards_balance | rewards balance | cleaning |
| clinc150 | pay_bill | paying a bill | grammatical |
| clinc150 | spending_history | spending history | cleaning |
| clinc150 | pto_request_status | pto request status | cleaning |
| clinc150 | credit_score | credit score | cleaning |
| clinc150 | new_card | new card | cleaning |
| clinc150 | lost_luggage | lost luggage | cleaning |
| clinc150 | oil_change_when | when to change oil | grammatical |
| clinc150 | yes | assertion | grammatical |
| clinc150 | travel_suggestion | travel suggestion | cleaning |
| clinc150 | todo_list_update | todo list update | cleaning |
| clinc150 | change_speed | changing speed | grammatical |
| clinc150 | tire_pressure | tire pressure | cleaning |
| clinc150 | no | negation | grammatical |
| clinc150 | nutrition_info | nutrition info | cleaning |
| clinc150 | carry_on | carry ons | grammatical |
| clinc150 | pto_used | pto used | cleaning |
| clinc150 | schedule_maintenance | scheduling maintenance | grammatical |
| clinc150 | travel_notification | travel notification | cleaning |
| clinc150 | sync_device | sync device | cleaning |
| clinc150 | thank_you | thank you | cleaning |
| clinc150 | roll_dice | roll dice | cleaning |
| clinc150 | food_last | food expiration | grammatical |
| clinc150 | cook_time | cook time | cleaning |
| clinc150 | reminder_update | reminder update | cleaning |
| clinc150 | report_lost_card | reporting a lost card | grammatical |

Table 12: Label cleaning.

| dataset | class name | rephrased class name | rephrased type |
|---------|-----------|---------------------|----------------|
| clinc150 | ingredient_substitution | ingredient substitution | cleaning |
| clinc150 | make_call | making a call | grammatical |
| clinc150 | todo_list | todo list | cleaning |
| clinc150 | change_accent | changing an accent | grammatical |
| clinc150 | bill_due | bill due | cleaning |
| clinc150 | damaged_card | damaged card | cleaning |
| clinc150 | restaurant_reviews | restaurant reviews | cleaning |
| clinc150 | do_you_have_pets | a question about your pets | grammatical |
| clinc150 | schedule_meeting | scheduling a meeting | grammatical |
| clinc150 | gas_type | gas type | cleaning |
| clinc150 | plug_type | plug type | cleaning |
| clinc150 | tire_change | tire change | cleaning |
| clinc150 | exchange_rate | exchange rate | cleaning |
| clinc150 | next_holiday | next holiday | cleaning |
| clinc150 | change_volume | changing volume | grammatical |
| clinc150 | who_do_you_work_for | whom do you work for | grammatical |
| clinc150 | credit_limit | credit limit | cleaning |
| clinc150 | how_busy | waiting time | grammatical |
| clinc150 | accept_reservations | accepting reservations | grammatical |
| clinc150 | order_status | order status | cleaning |
| clinc150 | pin_change | pin change | cleaning |
| clinc150 | account_blocked | account blocked | cleaning |
| clinc150 | what_song | what song | cleaning |
| clinc150 | international_fees | international fees | cleaning |
| clinc150 | last_maintenance | last maintenance | cleaning |
| clinc150 | meeting_schedule | meeting schedule | cleaning |
| clinc150 | ingredients_list | ingredients list | cleaning |
| clinc150 | report_fraud | reporting fraud | grammatical |
| clinc150 | measurement_conversion | measurement conversion | cleaning |
| clinc150 | smart_home | smart home | cleaning |
| clinc150 | book_hotel | booking a hotel | grammatical |
| clinc150 | current_location | current location | cleaning |
| clinc150 | min_payment | min payment | cleaning |
| clinc150 | whisper_mode | whisper mode | cleaning |
| clinc150 | canceling | cancel | grammatical |
| clinc150 | international_visa | international visa | cleaning |
| clinc150 | pto_balance | pto balance | cleaning |
| clinc150 | reset_settings | reset settings | cleaning |
| clinc150 | what_is_your_name | what your name is | grammatical |
| clinc150 | direct_deposit | direct deposit | cleaning |
| clinc150 | interest_rate | interest rate | cleaning |
| clinc150 | credit_limit_change | credit limit change | cleaning |
| clinc150 | what_are_your_hobbies | what your hobbies are | grammatical |

Table 13: Label cleaning.

| dataset | class name | rephrased class name | rephrased type |
|---|---|---|---|
| clinc150 | book_flight | booking a flight | grammatical |
| clinc150 | shopping_list | shopping list | cleaning |
| clinc150 | bill_balance | bill balance | cleaning |
| clinc150 | share_location | sharing location | grammatical |
| clinc150 | redeem_rewards | redeem rewards | cleaning |
| clinc150 | play_music | asking to play music | grammatical |
| clinc150 | calendar_update | calendar update | cleaning |
| clinc150 | are_you_a_bot | asking if you are a bot | grammatical |
| clinc150 | expiration_date | expiration date | cleaning |
| clinc150 | update_playlist | updating playlist | grammatical |
| clinc150 | cancel_reservation | canceling reservation | grammatical |
| clinc150 | tell_joke | telling a joke | grammatical |
| clinc150 | change_ai_name | changing ai name | grammatical |
| clinc150 | how_old_are_you | how old you are | grammatical |
| clinc150 | car_rental | car rental | cleaning |
| clinc150 | jump_start | jump start | cleaning |
| clinc150 | meal_suggestion | meal suggestion | cleaning |
| clinc150 | order_checks | ordering checks | grammatical |
| clinc150 | card_declined | a declined card | grammatical |
| cuad | Filename | | deletion |
| cuad | Document Name | | deletion |
| cuad | Document Name-Answer | | deletion |
| cuad | Parties | | deletion |
| cuad | Parties-Answer | | deletion |
| cuad | Agreement Date | | deletion |
| cuad | Agreement Date-Answer | | deletion |
| cuad | Effective Date | | deletion |
| cuad | Effective Date-Answer | | deletion |
| cuad | Expiration Date-Answer | | deletion |
| cuad | Renewal Term-Answer | | deletion |
| cuad | Notice Period To Terminate Renewal- Answer | | deletion |
| cuad | Governing Law-Answer | | deletion |
| cuad | Most Favored Nation-Answer | | deletion |
| cuad | Competitive Restriction Exception-Answer | | deletion |
| cuad | Non-Compete-Answer | | deletion |
| cuad | Exclusivity-Answer | | deletion |
| cuad | No-Solicit Of Customers-Answer | | deletion |
| cuad | No-Solicit Of Employees-Answer | | deletion |
| cuad | Non-Disparagement-Answer | | deletion |
| cuad | Termination For Convenience-Answer | | deletion |

Table 14: Label cleaning.

| dataset | class name | rephrased class name | rephrased type |
|---|---|---|---|
| cuad | Rofr/Rofo/Rofn-Answer | | deletion |
| cuad | Change Of Control-Answer | | deletion |
| cuad | Anti-Assignment-Answer | | deletion |
| cuad | Revenue/Profit Sharing-Answer | | deletion |
| cuad | Price Restrictions-Answer | | deletion |
| cuad | Minimum Commitment-Answer | | deletion |
| cuad | Volume Restriction-Answer | | deletion |
| cuad | Ip Ownership Assignment-Answer | | deletion |
| cuad | Joint Ip Ownership-Answer | | deletion |
| cuad | License Grant-Answer | | deletion |
| cuad | Non-Transferable License-Answer | | deletion |
| cuad | Affiliate License-Licensor-Answer | | deletion |
| cuad | Affiliate License-Licensee-Answer | | deletion |
| cuad | Unlimited/All-You-Can-Eat-License-Answer | | deletion |
| cuad | Irrevocable Or Perpetual License-Answer | | deletion |
| cuad | Source Code Escrow-Answer | | deletion |
| cuad | Post-Termination Services-Answer | | deletion |
| cuad | Audit Rights-Answer | | deletion |
| cuad | Uncapped Liability-Answer | | deletion |
| cuad | Cap On Liability-Answer | | deletion |

Table 15: Label cleaning.

| dataset | class name | rephrased class name | rephrased type |
|---|---|---|---|
| cuad | Liquidated Damages-Answer | | deletion |
| cuad | Warranty Duration-Answer | | deletion |
| cuad | Insurance-Answer | | deletion |
| cuad | Covenant Not To Sue-Answer | | deletion |
| cuad | Third Party Beneficiary-Answer | | deletion |
| 20 newsgroup | alt atheism | atheism | readability |
| 20 newsgroup | comp graphics | computer graphics | readability |
| 20 newsgroup | comp os ms-windows misc | microsoft windows | readability |
| 20 newsgroup | comp sys ibm pc hardware | pc hardware | readability |
| 20 newsgroup | comp sys mac hardware | mac hardware | readability |
| 20 newsgroup | comp windows x | windows x | readability |
| 20 newsgroup | misc forsale | for sale | readability |
| 20 newsgroup | rec autos | cars | readability |
| 20 newsgroup | rec motorcycles | motorcycles | readability |
| 20 newsgroup | rec sport baseball | baseball | readability |
| 20 newsgroup | rec sport hockey | hockey | readability |
| 20 newsgroup | sci crypt | cryptography | readability |
| 20 newsgroup | sci electronics | electronics | readability |
| 20 newsgroup | sci med | medicine | readability |
| 20 newsgroup | sci space | space | readability |
| 20 newsgroup | soc religion christian | christianity | readability |
| 20 newsgroup | talk politics guns | guns | readability |
| 20 newsgroup | talk politics mideast | middle east | readability |
| 20 newsgroup | talk politics misc | politics | readability |
| 20 newsgroup | talk religion misc | religion | readability |
| banking77 | activate my card | activating my card | grammatical |
| banking77 | age limit | age limit | cleaning |
| banking77 | apple pay or google pay | apple pay or google pay | cleaning |
| banking77 | atm support | atm support | cleaning |
| banking77 | automatic top up | automatic top up | cleaning |
| banking77 | balance not updated after bank transfer | balance that has not been updated after a bank transfer | grammatical |
| banking77 | balance not updated after cheque or cash deposit | balance that has not been updated after cheque or cash deposit | grammatical |

Table 16: Label cleaning.

| dataset | class name | rephrased class name | rephrased type |
|---|---|---|---|
| banking77 | beneficiary not allowed | a beneficiary who is not allowed | grammatical |
| banking77 | cancel transfer | canceling a transfer | grammatical |
| banking77 | card about to expire | a card that is about to expire | grammatical |
| banking77 | card acceptance | card acceptance | cleaning |
| banking77 | card arrival | card arrival | cleaning |
| banking77 | card delivery estimate | card delivery estimation | grammatical |
| banking77 | card linking | card linking | cleaning |
| banking77 | card not working | card not working | cleaning |
| banking77 | card payment fee charged | a card payment fee that was charged | grammatical |
| banking77 | card payment not recognised | card payment not recognised | cleaning |
| banking77 | card payment wrong exchange rate | card payment wrong exchange rate | cleaning |
| banking77 | card swallowed | card swallowed | cleaning |
| banking77 | cash withdrawal charge | cash withdrawal charge | cleaning |
| banking77 | cash withdrawal not recognised | cash withdrawal not recognised | cleaning |
| banking77 | change pin | changing pin | grammatical |
| banking77 | compromised card | compromised card | cleaning |
| banking77 | contactless not working | contactless not working | cleaning |
| banking77 | country support | country support | cleaning |
| banking77 | declined card payment | declined card payment | cleaning |
| banking77 | declined cash withdrawal | a declined cash withdrawal | grammatical |
| banking77 | declined transfer | declined transfer | cleaning |
| banking77 | direct debit payment not recognised | direct debit payment not recognised | cleaning |
| banking77 | disposable card limits | disposable card limits | cleaning |
| banking77 | edit personal details | editing personal details | grammatical |
| banking77 | exchange charge | exchange charge | cleaning |
| banking77 | exchange rate | exchange rate | cleaning |
| banking77 | exchange via app | exchange via app | cleaning |
| banking77 | extra charge on statement | extra charge on statement | cleaning |
| banking77 | failed transfer | failed transfer | cleaning |
| banking77 | fiat currency support | fiat currency support | cleaning |
| banking77 | get disposable virtual card | getting disposable virtual card | grammatical |
| banking77 | get physical card | getting physical card | grammatical |
| banking77 | getting spare card | getting spare card | cleaning |
| banking77 | getting virtual card | getting virtual card | cleaning |
| banking77 | lost or stolen card | lost or stolen card | cleaning |
| banking77 | lost or stolen phone | lost or stolen phone | cleaning |

Table 17: Label cleaning.

| dataset | class name | rephrased class name | rephrased type |
|---|---|---|---|
| banking77 | order physical card | ordering physical card | grammatical |
| banking77 | passcode forgotten | forgotten passcode | grammatical |
| banking77 | pending card payment | pending card payment | cleaning |
| banking77 | pending cash withdrawal | pending cash withdrawal | cleaning |
| banking77 | pending top up | pending top up | cleaning |
| banking77 | pending transfer | pending transfer | cleaning |
| banking77 | pin blocked | blocked pin | grammatical |
| banking77 | receiving money | receiving money | cleaning |
| banking77 | Refund not showing up | refund not showing up | cleaning |
| banking77 | request refund | refund request | grammatical |
| banking77 | reverted card payment? | reverted card payment | grammatical |
| banking77 | supported cards and curren-cies | supported cards and currencies | cleaning |
| banking77 | terminate account | terminating account | grammatical |
| banking77 | top up by bank transfer charge | top up by bank transfer charge | cleaning |
| banking77 | top up by card charge | top up by card charge | cleaning |
| banking77 | top up by cash or cheque | top up by cash or cheque | cleaning |
| banking77 | top up failed | failed top up | grammatical |
| banking77 | top up limits | top up limits | cleaning |
| banking77 | top up reverted | top up reverted | cleaning |
| banking77 | topping up by card | topping up by card | cleaning |
| banking77 | transaction charged twice | transaction charged twice | cleaning |
| banking77 | transfer fee charged | charged transfer fee | grammatical |
| banking77 | transfer into account | transferring into account | grammatical |
| banking77 | transfer not received by re-cipient | transfer not received by recipi-ent | cleaning |
| banking77 | transfer timing | transfer timing | cleaning |
| banking77 | unable to verify identity | being unable to verify identity | grammatical |
| banking77 | verify my identity | verifying my identity | grammatical |
| banking77 | verify source of funds | verifying source of funds | grammatical |
| banking77 | verify top up | verifying top up | grammatical |

Table 18: Label cleaning.

| dataset | class name | rephrased class name | rephrased type |
|---------|-----------|---------------------|----------------|
| banking77 | virtual card not working | virtual card not working | cleaning |
| banking77 | visa or mastercard | visa or mastercard | cleaning |
| banking77 | why verify identity | why identity verification is necessary | grammatical |
| banking77 | wrong amount of cash received | wrong amount of cash received | cleaning |
| banking77 | wrong exchange rate for cash withdrawal | wrong exchange rate for cash withdrawal | cleaning |
| dbpedia | Artist | artist | lower case |
| dbpedia | Plant | plant | lower case |
| dbpedia | Album | album | lower case |
| dbpedia | Animal | animal | lower case |
| dbpedia | Mean-Of-Transportation | mean of transportation | readability |
| dbpedia | NaturalPlace | natural place | readability |
| dbpedia | Athlete | athlete | lower case |
| dbpedia | OfficeHolder | office holder | readability |
| dbpedia | Company | company | lower case |
| dbpedia | Film | film | lower case |
| dbpedia | Educational-Institution | educational institution | readability |
| dbpedia | WrittenWork | written work | readability |
| dbpedia | Building | building | lower case |
| dbpedia | Village | village | lower case |
| ledgar | general | | deletion |
| law stack exchange | contract-law | contract law | cleaning |
| law stack exchange | constitutional-law | constitutional law | cleaning |
| law stack exchange | criminal-law | criminal law | cleaning |
| law stack exchange | tax-law | tax law | cleaning |
| law stack exchange | civil-law | civil law | cleaning |
| law stack exchange | intellectual-property | intellectual property | cleaning |
| massive | datetime query | getting date or time details | grammatical |
| massive | iot hue lightchange | changing hue light | grammatical |
| massive | transport ticket | getting a transport ticket | grammatical |
| massive | takeaway query | getting a takeaway | grammatical |
| massive | qa stock | stock | grammatical |
| massive | general greet | greeting | grammatical |
| massive | recommendation events | event recommendation | grammatical |
| massive | music dislikeness | music dislikeness | cleaning |
| massive | iot wemo off | turning off wemo | grammatical |

Table 19: Label cleaning.

| dataset | class name | rephrased class name | rephrased type |
|---------|-----------|---------------------|----------------|
| massive | cooking recipe | cooking recipes | grammatical |
| massive | qa currency | currency | grammatical |
| massive | transport traffic | transport traffic | cleaning |
| massive | general quirky | quirky issues | grammatical |
| massive | weather query | the weather | grammatical |
| massive | audio volume up | turning up the volume | grammatical |
| massive | email addcontact | adding email contact | grammatical |
| massive | takeaway order | a takeaway order | grammatical |
| massive | email querycontact | getting email contact | grammatical |
| massive | iot hue lightup | increasing hue light | grammatical |
| massive | recommendation locations | location recommendations | grammatical |
| massive | play audiobook | playing an audio book | grammatical |
| massive | lists createoradd | creating or adding lists | grammatical |
| massive | news query | the news | grammatical |
| massive | alarm query | getting alarm details | grammatical |
| massive | iot wemo on | turning wemo on | grammatical |
| massive | general joke | a joke | grammatical |
| massive | qa definition | definitions | grammatical |
| massive | social query | social media | grammatical |
| massive | music settings | music settings | cleaning |
| massive | audio volume other | audio volume | grammatical |
| massive | calendar remove | removing from calendar | grammatical |
| massive | iot hue lightdim | dimming hue light | grammatical |
| massive | calendar query | getting calendar details | grammatical |
| massive | email sendemail | sending en email | grammatical |
| massive | iot cleaning | cleaning | grammatical |
| massive | audio volume down | turning down volume | grammatical |
| massive | play radio | playing the radio | grammatical |
| massive | cooking query | cooking details | grammatical |
| massive | datetime convert | converting date or time | grammatical |
| massive | qa maths | math | grammatical |
| massive | iot hue lightoff | turning off hue light | grammatical |
| massive | iot hue lighton | turning on hue light | grammatical |
| massive | transport query | getting transport details | grammatical |
| massive | music likeness | music likeness | cleaning |
| massive | email query | getting email details | grammatical |
| massive | play music | playing music | grammatical |
| massive | audio volume mute | muting audio volume | grammatical |
| massive | social post | posting on social media | grammatical |
| massive | alarm set | setting an alarm | grammatical |
| massive | qa factoid | factoids | grammatical |
| massive | calendar set | setting the calendar | grammatical |

Table 20: Label cleaning.

| dataset | class name | rephrased class name | rephrased type |
|---|---|---|---|
| massive | play game | playing a game | grammatical |
| massive | alarm remove | removing an alarm | grammatical |
| massive | lists remove | removing from lists | grammatical |
| massive | transport taxi | transport taxi | cleaning |
| massive | recommendation movies | movie recommendations | grammatical |
| massive | iot coffee | making coffee | grammatical |
| massive | music query | getting music details | grammatical |
| massive | play podcasts | playing podcasts | grammatical |
| massive | lists query | getting lists details | grammatical |
| ag news | World | world | lower case |
| ag news | Sci/Tech | science and technology | readability |
| ag news | Sports | sports | lower case |
| ag news | Business | business | lower case |
| yahoo answers | Sports | sports | lower case |
| yahoo answers | Health | health | lower case |
| yahoo answers | Family & Relationships | family and relationships | readability |
| yahoo answers | Science & Mathematics | science and mathematics | readability |
| yahoo answers | Education & Reference | education and reference | readability |
| yahoo answers | Entertainment & Music | entertainment and music | readability |
| yahoo answers | Society & Culture | society and culture | readability |
| yahoo answers | Business & Finance | business and finance‚Äô | readability |
| yahoo answers | Politics & Government | politics and government | readability |
| yahoo answers | Computers & Internet | computers and internet | readability |
| xglue | foodanddrink | food and drink | readability |
| sentivent | cs r/brand | brand | readability |
| reuters | pet-chem | | deletion |
| reuters | income | | deletion |
| reuters | strategic-metal | | deletion |
| reuters | lei | | deletion |
| reuters | rand | | deletion |
| reuters | coconut-oil | | deletion |
| reuters | nkr | | deletion |
| reuters | oat | | deletion |
| reuters | propane | | deletion |
| reuters | saudriyal | | deletion |

Table 21: Label cleaning.

| dataset | class name | rephrased class name | rephrased type |
|---|---|---|---|
| reuters | sorghum | | deletion |
| reuters | tea | | deletion |
| reuters | cotton-oil | | deletion |
| reuters | nat-gas | nat gas | cleaning |
| reuters | fuel | | deletion |
| reuters | citruspulp | | deletion |
| reuters | nzdlr | | deletion |
| reuters | stg | | deletion |
| reuters | sun-meal | | deletion |
| reuters | cruzado | | deletion |
| reuters | dfl | | deletion |
| reuters | castorseed | | deletion |
| reuters | rice | | deletion |
| reuters | cornglutenfeed | | deletion |
| reuters | cpi | | deletion |
| reuters | meal-feed | | deletion |
| reuters | gnp | gross national product | readability |
| reuters | ship | ships | readability |
| reuters | acq | acquisition | readability |
| reuters | barley | | deletion |
| reuters | lin-oil | | deletion |
| reuters | corn-oil | | deletion |
| reuters | silver | | deletion |
| reuters | soy-meal | | deletion |
| reuters | tapioca | | deletion |
| reuters | orange | | deletion |
| reuters | plywood | | deletion |
| reuters | lead | | deletion |
| reuters | tin | | deletion |

Table 22: Label cleaning.

| dataset | class name | rephrased class name | rephrased type |
|---------|------------|----------------------|----------------|
| reuters | f-cattle | f cattle | cleaning |
| reuters | linseed | | deletion |
| reuters | pork-belly | | deletion |
| reuters | jobs | | deletion |
| reuters | naphtha | | deletion |
| reuters | rye | | deletion |
| reuters | lumber | | deletion |
| reuters | dkr | | deletion |
| reuters | platinum | | deletion |
| reuters | money-supply | money supply | cleaning |
| reuters | rape-meal | | deletion |
| reuters | coconut | | deletion |
| reuters | gas | | deletion |
| reuters | heat | | deletion |
| reuters | retail | | deletion |
| reuters | ipi | | deletion |
| reuters | ringgit | | deletion |
| reuters | copra-cake | | deletion |
| reuters | zinc | | deletion |
| reuters | rapeseed | | deletion |
| reuters | cpu | | deletion |
| reuters | fishmeal | | deletion |
| reuters | soy-oil | | deletion |
| reuters | veg-oil | vegetable oil | readability |
| reuters | yen | | deletion |
| reuters | carcass | | deletion |
| reuters | lin-meal | | deletion |
| reuters | red-bean | | deletion |
| reuters | jet | | deletion |
| reuters | wpi | | deletion |
| reuters | castor-oil | | deletion |
| reuters | copper | | deletion |
| reuters | wool | | deletion |
| reuters | cocoa | | deletion |
| reuters | groundnut-oil | | deletion |
| reuters | peseta | | deletion |
| reuters | palm-oil | | deletion |
| reuters | dmk | | deletion |
| reuters | bop | | deletion |
| reuters | l-cattle | | deletion |
| reuters | instal-debt | | deletion |
| reuters | iron-steel | | deletion |
| reuters | reserves | | deletion |
| reuters | rubber | | deletion |
| reuters | rape-oil | | deletion |
| reuters | housing | | deletion |
| reuters | inventories | | deletion |
| reuters | potato | | deletion |
| reuters | hog | | deletion |
| reuters | earn | earnings | readability |
| reuters | skr | | deletion |
| reuters | sun-oil | | deletion |
| reuters | palladium | | deletion |
| reuters | sunseed | | deletion |
| reuters | can | | deletion |
| reuters | soybean | | deletion |
| reuters | cotton | | deletion |
| reuters | austdlr | | deletion |
| reuters | palmkernel | | deletion |
| reuters | groundnut | | deletion |
| reuters | alum | | deletion |
| reuters | money-fx | money foreign exchange | readability |
| reuters | nickel | | deletion |

Table 23: Label cleaning.