# OpenReview forum: "Zero-shot Topical Text Classification with LLMs - an Experimental Study"
_EMNLP/2023/Conference — EMNLP 2023 Findings_

### Official Review · Reviewer_aRE1 · 2023-07-23

**Typos Grammar Style And Presentation Improvements:** N/A
**Soundness:** 3

**Excitement:**

2: Mediocre: This paper makes marginal contributions (vs non-contemporaneous work), so I would rather not see it in the conference.

**Missing References:**

N/A

**Paper Topic And Main Contributions:**

The paper presents a systematic study of the performance of large language models on zero-shot topical text classification (TTC) over a diverse benchmark of 23 datasets (TTC23). In this paper, the author compares several approaches to zero-shot TTC including similarity-based models, QA-based models, NLI-based models, and instruction-tuned models. It also proposes training TTC models by fine-tuning on existing labeled TTC datasets. The proposed TTC23 dataset includes 23 existing datasets covering diverse domains and styles. The paper preprocesses them into a unified format and cleanses labels for consistency. The paper conducts experiments comparing aforementioned methods on TTC23 and found out fine-tuning on TTC23 significantly improves zero-shot performance over using off-the-shelf models. Among the LLMs tried, Flan-T5-XXL FT obtains the top results. The author concludes LLMs exhibit strong zero-shot TTC capabilities, and fine-tuning on diverse TTC data can further enhance performance on new unseen categories.

**Questions For The Authors:**

- Given the trend of not open-sourcing the dataset used to train LLMs, would the proposed dataset and evaluation method applicable to good amount of LLMs?

**Reasons To Accept:**

- This paper presents the first comprehensive benchmark (TTC23) to evaluate zero-shot topical text classification, covering a diverse range of datasets. The proposed dataset and evaluation approach will facilitate future research.

- This paper provides an extensive empirical comparison of various methods for zero-shot TTC. The paper demonstrates that fine-tuning large LMs like Flan-T5 on TTC data significantly improves their zero-shot performance on unseen topics.

- This paper also presents experiments on the effect of label semantics, and inference speed tradeoffs, providing useful insights.

**Reasons To Reject:**

- This paper does not evaluate powerful recent LLMs like ChatGPT, over the Llama Families. While the author argues the dataset used to train those models are unclear, it is not convincing not trying them (Especially given the training data for T5 base model is also unclear). Given the missed SOTA LLMs could be competitive or superior for zero-shot TTC, this Limits the generality of conclusions.

- The largest model tries in this paper contains 13B parameters, which is still rather a small one without much "emergence" ability. With powerful LLMs, zero-shot or ICL using selective examples from the training subset could be an even cheaper approach as comparing to fine-tuning, which is missing in the paper's experiments or discussion.



**Reproducibility:**

4: Could mostly reproduce the results, but there may be some variation because of sample variance or minor variations in their interpretation of the protocol or method.

**Reviewer Confidence:**

4: Quite sure. I tried to check the important points carefully. It's unlikely, though conceivable, that I missed something that should affect my ratings.

---

> ### Author Rebuttal · Authors · 2023-08-28
>
> We appreciate the reviewer's insights.
>
> On the topic of LLMs, several challenges arise when evaluating them in our context.
>
> Firstly, the training methodology of chatGPT/GPT4 remains undisclosed. Thus, we can’t know if any of the datasets evaluated in our paper were used for supervised training by the GPT models. This is different from flan-t5 / t5  where the datasets used for flan's training are identified (https://github.com/google-research/FLAN/blob/main/flan/v2/flan_collection_info.csv), and t5's pre-training data is public and unsupervised (https://huggingface.co/datasets/c4).
>
> Secondly, accessing these models through a service poses reproducibility issues if the service evolves in the future. The associated financial implications of using this paid service, especially given our paper's scale, further complicate matters. As for the Llama models, our initial attempts with Llama in zero-shot scenarios underperformed, leading to its omission. These nuances will be elaborated upon in the revised manuscript.
>
> Furthermore, our results indicates that the benefits of fine-tuning on topical datasets span various model sizes, ranging from 355M to 11B. This implies that the trend may continue for larger models.
>
> In-context learning is a viable route. Yet, from a user's viewpoint, once fine-tuning is executed on a separate dataset from the test data, the distinction between a fine-tuned model and an original becomes imperceptible, leading to consistent inference efforts and costs. Conversely, in-context learning demands user intervention in curating few-shot examples. Hence, we chose to spotlight the zero-shot paradigm, given its potential appeal in real-world applications
>
> Regarding the trend mentioned by the reviewer, of not open-sourcing the datasets used to train LLMs  - while we concur that such a tendency is emerging, we remain optimistic about the sustained demand for open-source datasets and the associated LLMs, that will maintain the relevance and importance of the results and paradigms presented in our work.

---

### Official Review · Reviewer_UZow · 2023-08-05

**Soundness:** 3

**Excitement:**

4: Strong: This paper deepens the understanding of some phenomenon or lowers the barriers to an existing research direction.

**Missing References:**

-

**Paper Topic And Main Contributions:**

The paper focuses on Topical Text Classification (TTC), an established research area within natural language processing. The authors address the question of how well large language models (LLMs) can perform in a zero-shot scenario for TTC, considering the recent advancements in these models. They introduce a comprehensive study that evaluates the zero-shot performance of various LLMs on a benchmark called TTC23 , which consists of 23 publicly available TTC datasets spanning diverse domains and styles.

Through their benchmark and experiments, the authors illustrate the potential of specialized LLMs and provide insights into the practical use of these models for TTC tasks. The work contributes to advancing the understanding of the capabilities and limitations of LLMs in zero-shot TTC scenarios, providing guidance for practitioners interested in this field.

**Questions For The Authors:**

A. How fair is the comparison of the models? Is there a way to conclude about the effect of data used during pre-training of the models at hand on their performance?

B. If you could provide more details about the decoding strategy of the models, it would be very helpful to see if there is an important effect of this configuration on their performance in TTC.

**Reasons To Accept:**

-The paper exhibits a high level of organization and clarity, effectively conveying the proposed methodology.
- Comprehensive experiments involving various pre-trained models are conducted across diverse datasets, contributing to the paper's robustness.
-The integration of tables, summaries, and graphs enhances the reader's comprehension of the paper's content.
-The model's focus on addressing Topical Text Classification (TTC) aligns with a pertinent challenge. The authors' intention to sharing the benchmark and models with the public is noted.
-This approach holds potential utility not only within the realm of NLP but also extends to other domains where zero-shot classification finds applicability.

**Reasons To Reject:**

- The data, code, fine-tuned models and methodology have not been made public in the current version of the paper. While the authors have expressed their intention to release these resources, this currently hinders the ability to verify the stated claims, results, and procedures, or to replicate the conducted experiments.
- Further elaboration on the hyperparameters would have enhanced the comprehension of the study's nuances. For instance, details about the decoding strategy, such as whether beam search or sampling techniques were employed, are lacking. This aspect could influence both the model's performance and the runtime during inference.
- The potential impact of the pre-training data on the models' performance in the TTC task received limited discussion. Considering whether certain models were previously exposed to similar texts or tasks during their pre-training could partially explain performance gaps between the models.
- An exploration of the models' strengths and weaknesses at the dataset level would have been valuable. The addition of metrics like standard deviation, minimum, and maximum values might have revealed patterns that could guide the scope of future research approaches

**Reproducibility:**

4: Could mostly reproduce the results, but there may be some variation because of sample variance or minor variations in their interpretation of the protocol or method.

**Reviewer Confidence:**

3: Pretty sure, but there's a chance I missed something. Although I have a good feel for this area in general, I did not carefully check the paper's details, e.g., the math, experimental design, or novelty.

**Typos Grammar Style And Presentation Improvements:**

-

---

> ### Author Rebuttal · Authors · 2023-08-28
>
> We appreciate the reviewer's feedback.
>
> Regarding the potential reproducibility, as stated in the manuscript, upon acceptance, we commit to share the code, data, and models in their entirety, to ensure our flow is clearly reproduceable. We believe that this follows standard practice in the field.
>
> For the decoding strategy, we employed greedy generation. Given the objective of classification tasks is to identify the class with maximum model confidence, a greedy approach is apt. Factors like creativity and diversity, pivotal for generative tasks, are secondary here. We'll address this in the manuscript.
>
> Typically, hyper-parameters cited in our paper are borrowed from analogous hugging-face experiments. Though deeper examination of their influence is possible, our study currently focuses on other aspects.
>
> We, too, recognize the importance of data's role in pre-training / training versus testing. Our paper has made strides in this direction via: 1) Assessing the effect of class similarities in training and test data for fine-tuned folds (section 6.4); 2) Omitting datasets trained on by flan and the QA models; 3) Evaluating folds that distinctly differ between domains (section 6.3). A thorough analysis of data impact during LLMs' pre-training is intricate and was reserved due to space and time constraints.

---

### Official Review · Reviewer_PZ6f · 2023-08-06

**Soundness:** 4

**Excitement:**

2: Mediocre: This paper makes marginal contributions (vs non-contemporaneous work), so I would rather not see it in the conference.

**Missing References:**

Multitask Prompted Training Enables Zero-Shot Task Generalization. Victor Sanh, et al., ICLR 2022.

**Paper Topic And Main Contributions:**

The paper demonstrates that fine-tuning on multiple topical text classification datasets improves zero-shot classification, as tested with 11B LM.

**Questions For The Authors:**

N/A

**Reasons To Accept:**

The methodology is convincing, and the improvement (4% average over 23 datasets) is noticeable. New dataset is assembled.

**Reasons To Reject:**

Still the results are not surprising considering that this phenomenon was already observed in a more general (not limited to topical classification) context in the following paper: Multitask Prompted Training Enables Zero-Shot Task Generalization. Victor Sanh, et al., ICLR 2022, which should have been referenced here.

Also, in my observation, 11B model is not yet called large LM, which is normally reserved for 65B+ (starting with Llama and including GPT3+, Palm and a few others of that size).

Since the prompts are important in this context, an ablation study and sensitivity to the prompts could be have been explored in more depths.


**Reproducibility:**

4: Could mostly reproduce the results, but there may be some variation because of sample variance or minor variations in their interpretation of the protocol or method.

**Reviewer Confidence:**

4: Quite sure. I tried to check the important points carefully. It's unlikely, though conceivable, that I missed something that should affect my ratings.

**Typos Grammar Style And Presentation Improvements:**

No specific issues. But the content of the paper can be better suited for a journal or a short paper considering limited novelty of the main result.

---

> ### Author Rebuttal · Authors · 2023-08-28
>
> We thank the reviewer for thorough and constructive review.
>
> Thank you for highlighting the omitted reference to T0; we will incorporate it into our manuscript.
>
> In relation to the novelty of our research versus works like T0:
>
> The aim of most prior work in the field was to create a versatile zero-shot model that can generalize as much as possible, in the sense that it can perform well over a wide range of downstream tasks. We agree that previous work has shown that achieving such a zero-shot model is possible and thus that it is indeed not surprising that this can also be done on narrower scope of tasks like topical text classification.
>
> However, our aim here is different: we explore whether honing-in on a **narrower scope** of downstream tasks can enable us to **improve** the zero-shot performance on this type of task only.
>
> And since we focus on topical text classification, we propose to focus on fine-tuning the model on this type of tasks only using data from various domains. This enables us to significantly boost the zero-shot performance of the model on topical text classification, compared to the general-purpose models.
>
> To the best of our knowledge, this approach hasn’t been undertaken before, and it provides a valuable insight for the community, that can be further explored in future work for other tasks as well.
>
> That said, we concur that this distinction wasn't adequately stressed in the manuscript and will rectify this in our final version.
>
> In response to the comment on LLMs, we'll refine the terminology in our manuscript.
>
> However, we'd like to highlight a few challenges associated with evaluating 65B+ models.
>
> Firstly, assessing the zero-shot performance of the GPT family is challenging due to its undisclosed training data, potentially encompassing supervised training from our evaluation datasets. Secondly, given its paid nature, extensive inference at the paper's scale is not only costly but also non-reproducible should the service evolve in the future. Regarding the Llama family, our trials with Llama in zero-shot on TTC-23 yielded unsatisfactory results, leading to its exclusion. These points will be incorporated into the revised manuscript.
>
> As for the prompts, initial tests on the dev set affirmed their superiority over other prompts, including those for topical text classification from the flan collection. We'll detail these results upon the paper's acceptance.

---

### Meta-Review · Area_Chair_SLtx · 2023-09-22

**Recommendation:** 3

**Metareview:**

The paper proposes a benchmark for topical text classification called TTC23, which is curated from a diverse set of existing datasets for the task. The authors use this benchmark to explore the zero-shot capabilities of LLMs for the task, and also show that fine-tuning on diverse TTC data can further enhance performance on new unseen categories.

Reviewers agree that the proposed benchmark is comprehensive, and that the set of experiments presented is extensive, involving various pre-trained models. That will benefit "understanding the capabilities and limitations of LLMs in zero-shot TTC scenarios, providing guidance for practitioners interested in this field."

One of the concerns raised by the reviewers involves the novelty of the results, since similar conclusions have been obtained previously in more general settings. The authors argue that their contribution lays in providing in-depth insights in a narrower set of downstream tasks, which improves zero-shot performance in TTC in unseen ones. Another concern is regarding not comparing against ICL models, which the authors argue that, while relevant, does not align with their focus on allowing end users to only rely on zero-shot interactions with good performance. Authors are encouraged to include their valid justifications for not comparing against GPT models, as well as their clarifications for the effects of data used during pre-training and testing.

---

### Decision · Program_Chairs · 2023-10-07

**Decision:**

Accept-Findings

**Comment:**

The paper proposes a benchmark for topical text classification called TTC23, which is curated from a diverse set of existing datasets for the task. The authors use this benchmark to explore the zero-shot capabilities of LLMs for the task, and also show that fine-tuning on diverse TTC data can further enhance performance on new unseen categories.

Reviewers agree that the proposed benchmark is comprehensive, and that the set of experiments presented is extensive, involving various pre-trained models. That will benefit "understanding the capabilities and limitations of LLMs in zero-shot TTC scenarios, providing guidance for practitioners interested in this field."

One of the concerns raised by the reviewers involves the novelty of the results, since similar conclusions have been obtained previously in more general settings. The authors argue that their contribution lays in providing in-depth insights in a narrower set of downstream tasks, which improves zero-shot performance in TTC in unseen ones. Another concern is regarding not comparing against ICL models, which the authors argue that, while relevant, does not align with their focus on allowing end users to only rely on zero-shot interactions with good performance. Authors are encouraged to include their valid justifications for not comparing against GPT models, as well as their clarifications for the effects of data used during pre-training and testing.